



# Sources of low-frequency variability in observed Antarctic sea ice

David B. Bonan[1], Jakob Dörr[2,3], Robert C.J. Wills[4,5], Andrew F. Thompson[1], and Marius Årthun[2,3]

[1]Environmental Science and Engineering, California Institute of Technology, Pasadena, California, USA
[2]Geophysical Institute, University of Bergen, Bergen, Norway
[3]Bjerknes Centre for Climate Research, Bergen, Norway
[4]Department of Atmospheric Sciences, University of Washington, Seattle, Washington, USA
[5]Institute for Atmospheric and Climate Science, ETH Zurich, Zurich, Switzerland

**Correspondence:** David B. Bonan (dbonan@caltech.edu)

**Abstract.** Antarctic sea ice gradually increased from the late 1970s until 2016, when it experienced an abrupt decline. A number of mechanisms have been proposed for both the gradual increase and abrupt decline of Antarctic sea ice, but how each mechanism manifests spatially and temporally remains poorly understood. Here, we use a statistical method called low-frequency component analysis to analyze the spatial-temporal structure of observed Antarctic sea-ice concentration variability.

5    The identified patterns reveal distinct modes of low-frequency sea ice variability. The leading mode, which accounts for the large-scale, gradual expansion of sea ice, is associated with the Interdecadal Pacific Oscillation and resembles the observed sea-surface temperature trend pattern that climate models have trouble reproducing. The second mode is associated with the central Pacific El Niño–Southern Oscillation (ENSO) and the Southern Annular Mode, and accounts for most of the sea ice variability in the Ross Sea. The third mode is associated with the eastern Pacific ENSO and Amundsen Sea Low, and accounts for most of

10   the pan-Antarctic sea-ice variability and almost all of the sea ice variability in the Weddell Sea. This mode is associated with periods of abrupt Antarctic sea-ice decline and is related to a weakening of the circumpolar westerlies, which favors surface warming through a shoaling of the ocean mixed layer and decreased northward Ekman heat convergence. Broadly, these results suggest that climate model biases in long-term Antarctic sea-ice and global sea-surface temperature trends are related to each other and that eastern Pacific ENSO variability causes abrupt sea ice changes.



## 1  Introduction

Antarctic sea ice plays a crucial role in Earth's climate system. The seasonal cycle of Antarctic sea-ice cover, which expands and contracts by approximately 16 million km$^2$ each year, impacts the ocean's global overturning circulation through brine rejection and freshwater input (e.g., Abernathey et al., 2016; Pellichero et al., 2018). Antarctic sea-ice cover also exerts a strong control on Southern Ocean primary productivity (e.g., Arrigo et al., 1997; Lizotte, 2001; Arrigo and van Dijken, 2004; Smith and Comiso, 2008), carbon exchange (e.g., Fogwill et al., 2020), and low level clouds (e.g., DuVivier et al., 2021) by modulating air-sea heat, freshwater, and biogeochemical fluxes. Studies have also invoked Antarctic sea ice as a major player in glacial-interglacial cycles of the late Pleistocene through reorganization of the ocean's global overturning circulation (e.g., Keeling and Stephens, 2001; Ferrari et al., 2014; Marzocchi and Jansen, 2017). Understanding processes that contribute to trends and variability in observed Antarctic sea ice remains a central goal of climate science.

Since the late 1970s, Antarctic sea-ice area (SIA) has slowly increased, despite significant global warming (Fig. 1a; Parkinson and Cavalieri, 2012; Turner et al., 2015; Gagné et al., 2015; Parkinson, 2019). The increase in Antarctic SIA occurred largely between 2000 and 2014 (Fig. 1a; Gagné et al., 2015; Meehl et al., 2016) and was associated with increased sea-ice concentration in all sectors of the Antarctic, except for the Amundsen and Bellingshausen Seas (Fig. 1b). In 2016, Antarctic SIA experienced an abrupt decline that persisted until 2019 (Fig. 1a; Turner et al., 2017; Stuecker et al., 2017; Raphael and Handcock, 2022; Fogt et al., 2022). The abrupt decline in Antarctic sea-ice concentration occurred throughout much of the Antarctic, with some of the largest changes found in the Weddell Sea, Indian sector, and Ross Sea (Fig. 1c; Turner et al., 2017).

A number of mechanisms have been proposed for both the gradual expansion and abrupt decline of Antarctic sea ice. Meehl et al. (2016) argued that the gradual increase in Antarctic sea ice was caused by decadal climate variability emanating from the tropical Pacific that deepened the Amundsen Sea Low, strengthened the circumpolar westerlies, and caused surface cooling through enhanced Ekman heat flux divergence. Other studies argued that increased freshwater input, either from ice-shelf melt (Bintanja et al., 2013; Pauling et al., 2016; Sadai et al., 2020), changes in precipitation and evaporation (Fyfe et al., 2012; Purich et al., 2018) or sea ice melt itself (Haumann et al., 2020), can cause sea ice expansion by increasing subsurface stratification and preventing warmer, deeper waters from interacting with the surface. Antarctic sea-ice expansion has also been attributed to internal climate variability and variations in open ocean convection (Turner et al., 2016; Singh et al., 2019; Zhang et al., 2019). The abrupt decline of Antarctic sea ice, on the other hand, has been attributed to weakened circumpolar westerlies associated with intrinsic variability of the Southern Annular Mode (SAM), El Niño–Southern Oscillation (ENSO), and Indian Ocean Dipole (IOD; Stuecker et al., 2017; Schlosser et al., 2018; Wang et al., 2019; Purich and England, 2019). The abrupt decline of sea ice has also been attributed to a gradual build up of subsurface heat through ocean preconditioning (Meehl et al., 2019; Campbell et al., 2019; Zhang et al., 2022). Beyond the gradual expansion and abrupt decline of Antarctic sea ice, Antarctic sea ice also exhibits substantial interannual-to-decadal variability, which has been linked to the phasing of the SAM and ENSO (Thompson and Solomon, 2002; Fogt and Bromwich, 2006; Stammerjohn et al., 2008; Matear et al., 2015; Doddridge and



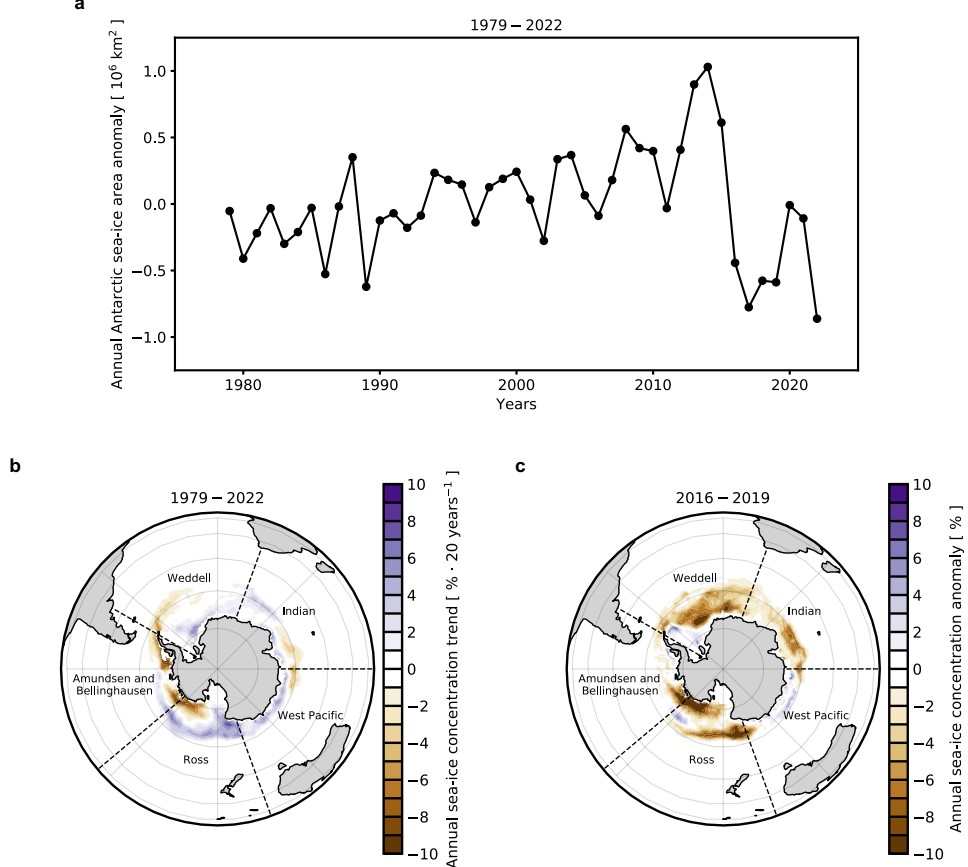

**Figure 1. Observed changes to Antarctic sea ice from 1979 to 2022.** (a) Anomalies in annual-mean Antarctic sea-ice area from 1979 to 2022 relative to the $1981-2010$ average. (b) Linear trend in annual-mean Antarctic sea-ice concentration from 1979 to 2022. (c) Anomalies in annual-mean Antarctic sea-ice concentration averaged from 2016 to 2019 relative to the $1981-2010$ average. Annual-mean Antarctic sea-ice area exhibits a slight positive trend of $0.15 \times 10^6$ km$^2 \cdot 40$ years$^{-1}$ from 1979 to 2022 and the SIA anomaly averaged from 2016 to 2019 is approximately $-0.60 \times 10^6$ km$^2$. The regions have longitude boundaries of $60°\text{W}-20°\text{E}$ (Weddell), $20°\text{E}-90°\text{E}$ (Indian), $90°\text{E}-160°\text{E}$ (West Pacific), $160°\text{E}-130°\text{W}$ (Ross), and $130°\text{W}-60°\text{W}$ (Amundsen and Bellingshausen), respectively.





Marshall, 2017; Holland et al., 2017; Crosta et al., 2021), zonal atmospheric wave structures (Raphael, 2007), surface wind
variability (Holland and Kwok, 2012), Pacific decadal variability (Chung et al., 2022) and Atlantic multidecadal variability
(Li et al., 2014; Eayrs et al., 2021). Indeed, a number of mechanisms can contribute to short- and long-term Antarctic sea-ice
variability, but a unified understanding of how each process manifests spatially and temporally in the observational record is
lacking.

This lack of understanding is, in part, because most coupled climate models — which oftentimes aid mechanistic understanding
of the climate system — have trouble reproducing the observed magnitude, sign, and spatial pattern of Antarctic sea-ice trends,
including periods of abrupt sea ice loss (Turner et al., 2013; Purich et al., 2016; Rosenblum and Eisenman, 2017; Roach et al.,
2020). Some studies have shown that internal climate variability may explain the disagreement in Antarctic sea-ice trends be-
tween climate models and observations (Polvani and Smith, 2013; Zunz et al., 2013; Mahlstein et al., 2013; Gagné et al., 2015;
Singh et al., 2019). Other studies have shown that climate models can reproduce the sign and magnitude of observed Antarctic
sea-ice trends if winds or sea ice motion are nudged to observed values (e.g., Blanchard-Wrigglesworth et al., 2021; Sun and
Eisenman, 2021), which suggests an important role for a strengthening of near-surface winds in Antarctic sea-ice expansion.
Indeed, ozone depletion is thought to cause a slight strengthening of the circumpolar westerlies (Thompson and Solomon,
2002), which over long timescales would favor surface cooling and promote sea ice growth through enhanced Ekman heat flux
divergence (Sigmond et al., 2011; Thompson et al., 2011; Kostov et al., 2017; Hartmann, 2022). But ozone depletion itself
has been shown to cause sea ice decline not sea ice expansion in climate models (Sigmond and Fyfe, 2010, 2014), which
complicates mechanistic understanding of the observed sea ice changes. Reconciling climate models and observations requires
a better understanding of the sources of sea ice trends and variability in the observational record.

In this study, we use low-frequency component analysis (LFCA; Wills et al., 2018; Schneider and Held, 2001), which iden-
tifies slowly evolving modes of variability, to examine low-frequency variability in observed annual-mean Antarctic sea-ice
concentration. LFCA makes no *a priori* assumptions about which processes contribute to low-frequency sea ice variability.
Additionally, while LFCA isolates low-frequency variability, it still retains high-frequency variability, enabling a robust char-
acterization of sea ice variability across timescales. In what follows, we first describe LFCA and the observational datasets
(Section 2). We then use LFCA to explore how different modes of variability have contributed to observed Antarctic sea-ice
changes (Section 3). Finally, we examine mechanisms for low-frequency Antarctic sea-ice variability (Section 4).

## 2  Data and methods

### 2.1  Observations

Estimates of monthly Antarctic sea-ice concentration were obtained from the NOAA/NSIDC Climate Data Record of passive
microwave sea ice concentration (Meier et al., 2013). Observation-based data of sea-surface temperatures (SSTs), 500 hPa
geopotential height (GPH), 10-m (near surface) zonal and meridional winds ($u_s$ and $v_s$), and the net surface heat flux, were





obtained from the ERA5 global reanalysis (Hersbach et al., 2020). Sparse data coverage of the Southern Ocean toward the beginning of the satellite era motivates the use of reanalysis data. We further use an estimate of the surface ocean mixed layer depth from the companion ORAS5 global ocean reanalysis (Zuo et al., 2019). While the choice of ocean reanalysis introduces errors into physical interpretations, we prefer reanalysis over direct observations for the former's spatial and temporal coverage. All sea ice and reanalysis data products are on monthly timescales from 1979 to 2022 and are used to compute annual means. The sea ice and reanalysis data products are regridded to a $1° \times 1°$ grid using second-order conservative remapping.

## 2.2 Low-frequency component analysis

LFCA is a statistical method (Wills et al., 2018; Schneider and Held, 2001) that finds the linear combination of empirical orthogonal functions (EOFs) that results in the highest ratio of low-frequency variance to total variance. We define low-frequency variance as the variance remaining after applying a 10-year cutoff low-pass filter, similar to previous studies that have applied LFCA to other climate variables (e.g., Wills et al., 2018; Årthun et al., 2021; Oldenburg et al., 2021; Dörr et al., 2023). LFCA has been used to characterize and understand modes of low-frequency variability in Atlantic and Pacific sea-surface temperature (Wills et al., 2019a, b; Årthun et al., 2021), Atlantic overturning circulation (Jiang et al., 2021), northward ocean heat transport (Oldenburg et al., 2021), and Arctic sea-ice concentration (Dörr et al., 2023), which is a companion study of this one.

In LFCA, the resulting anomaly patterns and timeseries are called low-frequency patterns (LFPs) and low-frequency components (LFCs), respectively (see Wills et al. (2018) for more details). In this study, the patterns and timeseries represent Antarctic sea-ice concentration anomalies relative to the $1981 - 2010$ average. The LFCs are normalized to have unit variance such that the LFPs show the anomaly pattern associated with a 1-standard-deviation anomaly in the corresponding LFC. The LFCs are required to be orthogonal (uncorrelated), but the LFPs are not. LFCA finds the pattern of variability within the included EOFs that has the maximum possible ratio of low-frequency to total variance and persistence, motivating its use over other statistical methods like dynamical adjustment (e.g., Smoliak et al., 2015) and principal component analysis (PCA). PCA, for example, takes advantage of the spatial structure of covariation in climate data to find a basis of EOFs that are ordered by the fraction of total variance they capture. Because of this, PCA maximizes the variance captured by the first EOF, and it can sometimes group together multiple processes and give spurious connections that are not rooted in shared physical mechanisms (e.g., Deser, 2000). LFCA instead identifies modes of variability based on their dominant time scale, providing a more physically consistent representation of variability in climate data.

The resulting LFPs are sorted by their ratio of low-frequency to total variance, which we refer to as the variance ratio ($r$). Here we retain the 5 leading EOFs, which accounts for approximately 70% of the total Antarctic sea-ice concentration variability. The choice of the number of EOFs to retain is subjective. We find that increasing the number of EOFs causes over-fitting issues that affects the physical interpretability, while decreasing the number EOFs to 3 accounts for less of the total sea ice





concentration variability and mixes modes of variability that appear to be physically distinct.

To quantify the effect of each LFC on Antarctic sea ice, the LFCs and LFPs are used to construct temporally evolving maps of Antarctic sea-ice concentration anomalies. This is done by multiplying each LFP by the corresponding LFC. The reconstructed maps are further used to calculate regional and pan-Antarctic SIA anomalies by multiplying the LFPs by the grid-cell area, and

summing up over the target region. This produces a timeseries of Antarctic SIA anomalies unique to each LFP. We also use the multi-taper spectral analysis method (Percival et al., 1993) to quantify the power of each LFC on specific timescales.

Improved isolation of the forced response in the leading pattern can be achieved by performing a combined analysis of sea ice concentration with other fields like sea surface temperature or sea level pressure. This was introduced by Wills et al. (2020)

using a signal-to-noise pattern recognition method (see also Bretherton et al., 1992). We performed a three-field analysis using SST and 500 hPa GPH and have determined that sea ice concentration alone is sufficient to isolate modes of low-frequency Antarctic sea-ice variability. For the multi-field analysis, SST and GPH anomaly matrices are concatenated with the sea ice concentration anomaly matrix in the spatial dimension. Each field variable is normalized by the trace of its covariance matrix such that all variables are unitless and weighted equally. The rest of the multi-field analysis proceeds exactly as in the individual

LFCA on sea ice concentration. Note, a companion study (i.e., Dörr et al., 2023) found that the combined analysis improved estimates of the forced component of Arctic sea-ice trends. We hypothesize that the multi-field approach works better for the Arctic because Arctic sea ice exhibits a stronger forced response associated with global warming than Antarctic sea ice does (e.g., Rosenblum and Eisenman, 2017).

## 3    Patterns of low-frequency Antarctic sea-ice variability

We begin by examining the LFCs and LFPs obtained by applying LFCA to annual-mean Antarctic sea-ice concentration from 1979 to 2022 with a 10-year cutoff low-pass filter and retaining the 5 leading EOFs (see Methods). Each LFP and LFC is spatially and temporally distinct, containing large regional structure in Antarctic sea-ice concentration and different characteristic timescales (Fig. 2). LFP1 exhibits a near pan-Antarctic wide signal of positive sea ice concentration anomalies with negative sea ice concentration anomalies in the Amundsen and Bellingshausen Seas, and it accounts for approximately

30% of the low-frequency variance (Fig. 2a). LFP1 is similar to the observed trend in Antarctic sea-ice concentration (compare Fig. 1a and Fig. 2a) and has a relatively high variance ratio ($r = 0.70$). The associated LFC exhibits a strong positive trend and a canonical red-noise spectra with increasing power at low-frequency ($> 10$ years) timescales. LFC1 also captures the increased positive trend in sea ice concentration between $2000 - 2014$ (Fig. 2a). LFP2 features a spatial pattern reminiscent of the SAM imprint on sea ice concentration (Lefebvre et al., 2004), with positive sea ice concentration anomalies in the Ross

Sea and negative sea ice concentration anomalies in the Weddell Sea (Fig. 2b). LFP2 accounts for approximately 21% of the low-frequency and has a smaller variance ratio ($r = 0.52$) than LFP1. The associated LFC exhibits strong positive values in the late 1990s and strong power at approximately 3- and 7- year timescales. LFP3 shows a large-scale pattern of positive sea





ice concentration anomalies mainly in and around the Weddell Sea (Fig. 2c). LFP3 accounts for approximately 12% of the low-frequency variance and has a variance ratio of $r = 0.38$. Notably, the associated LFC captures the abrupt decline in sea ice concentration that occurred around 2016. LFC3 also captures abrupt decline events in 1988 and 2010 and the persistent negative sea ice concentration anomalies seen since 2016. LFC3 exhibits strong power around 4- and 5- year timescales, suggesting an association with ENSO (e.g., Trenberth, 1997). Finally, LFP4 has strong negative sea ice concentration anomalies in the Amundsen and Bellingshausen Seas, with positive sea ice concentration anomalies in the Weddell Sea and Indian sector (Fig. 2d). LFP4 accounts for approximately 5% of the low-frequency variance and has a small variance ratio ($r = 0.17$). The associated LFC (LFC4) exhibits power on 3- and 4-year timescales. Figure A1 shows the last LFC and LFP, which essentially represents the residual from all higher frequency sea ice variability (see power spectra). LFP5 weakly resembles the spatial pattern of sea ice concentration anomalies associated with atmospheric zonal wave three as identified by Raphael (2007). LFC5 also shares some common features with the zonal wave three index (compare LFC5 with Fig. 7 of Raphael, 2007).

### 3.1 Contribution to sea ice concentration trends and variability

We next consider how each LFC contributes to trends and variability of observed Antarctic sea-ice concentration from 1979 to 2022 by projecting each LFC onto the corresponding LFP at each grid point. This produces a timeseries of sea ice concentration at each grid point that is unique to each LFC.

The dominant mode contributing to the gradual increase in Antarctic sea-ice concentration since the late 1970s is LFC1. A linear trend of the sea ice concentration associated with LFC1 shows large positive values throughout much of the Antarctic and weak negative values in the Amundsen and Bellingshausen Seas (Fig. 3a), which is consistent with the observed trend in sea ice concentration (Fig. 1a). The other three LFCs (Fig. 3b-d) contribute little to the long-term trend in Antarctic sea-ice concentration. However, LFC3 does contribute to a slight negative trend in sea ice concentration in and around the Weddell Sea (Fig. 3c), though these values are smaller than the large positive values seen in LFC1 (Fig. 3a).

While LFC1 dominates the long-term trend in sea ice concentration, it contributes little to the abrupt decline in 2016 and the years that followed (Fig. 4a). Antarctic sea-ice concentration anomalies from 2016 to 2019 are primarily related to LFC2-4 (Fig. 4b-d). LFC2 contributed to a decline in sea ice concentration in the Ross Sea, while LFC3 contributed to a decline in sea ice concentration mostly in the Weddell Sea, but also in other regions like the West Pacific sector and parts of the Ross Sea. LFC4 contributed some to the abrupt decline in sea ice concentration from 2016 to 2019, with small changes on the peripheral edge of the sea ice cover in the Weddell Sea and Indian sector (Fig. 4d).

### 3.2 Contribution to regional sea ice area changes

To better understand how each LFC contributes to the temporal evolution of Antarctic sea ice, we next examine the total and regional Antarctic SIA anomalies associated with each LFC and LFP (see Section 2.2). The regional domains broadly capture regions of distinct Antarctic sea-ice variability, as noted by Raphael and Hobbs (2014). These regions include: the Weddell



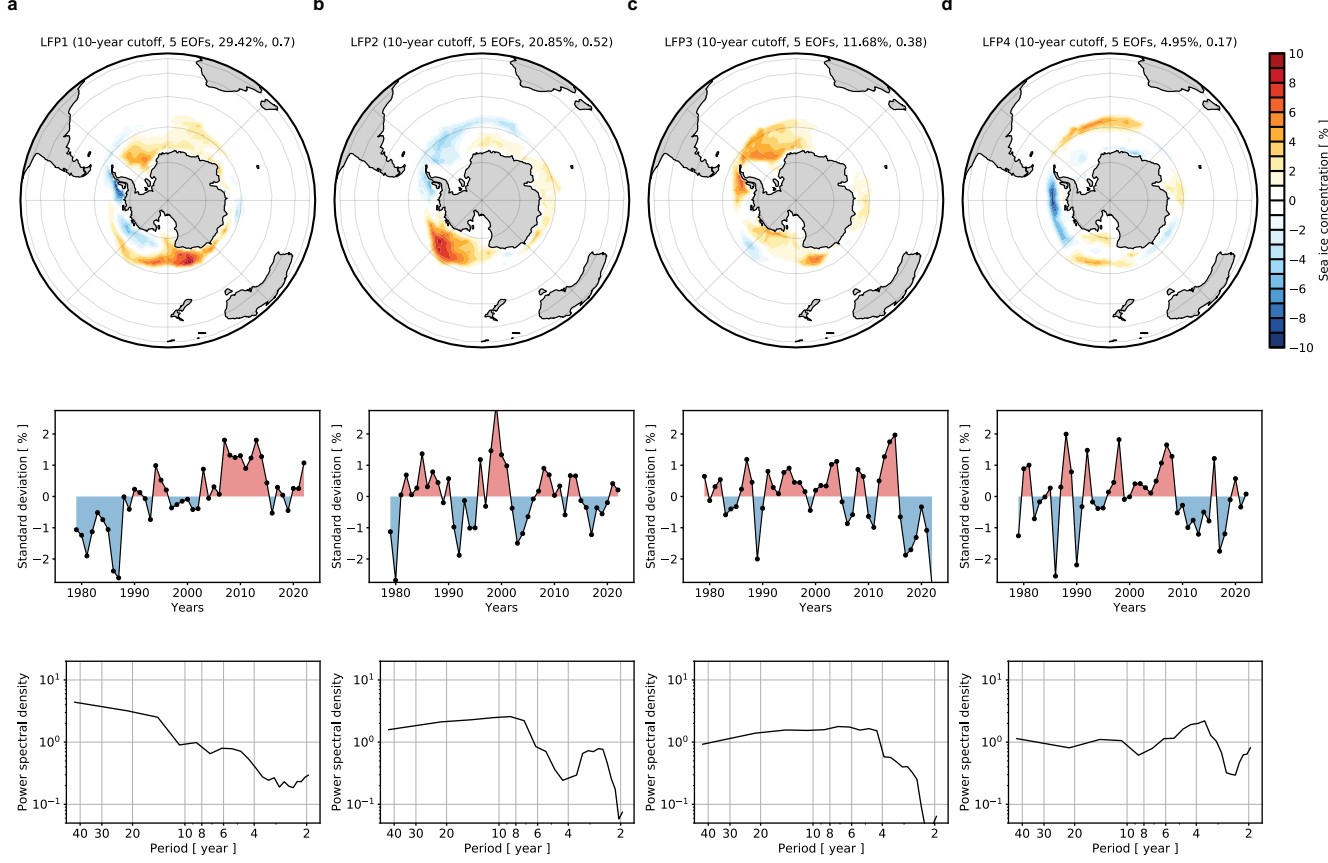

**Figure 2. Low-frequency variations of Antarctic sea-ice concentration.** (a) First, (b) second, (c) third, and (d) fourth (top) low-frequency patterns (LFPs), (middle) low-frequency components (LFCs), and (bottom) power spectra density of each LFC using a 10-year cutoff and retaining the five leading EOFs of annual-mean Antarctic sea-ice concentration anomalies from 1979 to 2022. Power spectra are computed with multitaper spectral analysis (Percival et al., 1993). The fraction of explained low-frequency variance (in %) and the ratio $r$ of low-frequency to total variance is given for each pattern.




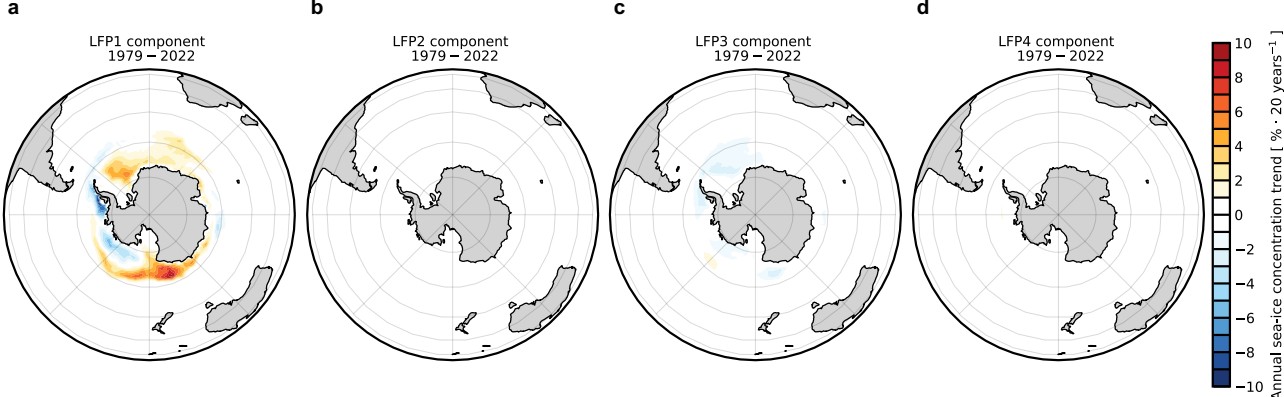

**Figure 3. Trends in low-frequency components of Antarctic sea-ice concentration.** Linear trend in annual-mean Antarctic sea-ice concentration anomalies from 1979 – 2022 associated with (a) LFC1, (b) LFC2, (c) LFC3, and (d) LFC4.

Sea, Indian sector, West Pacific sector, Ross Sea, and Amundsen and Bellingshausen Seas (see Figure 1).

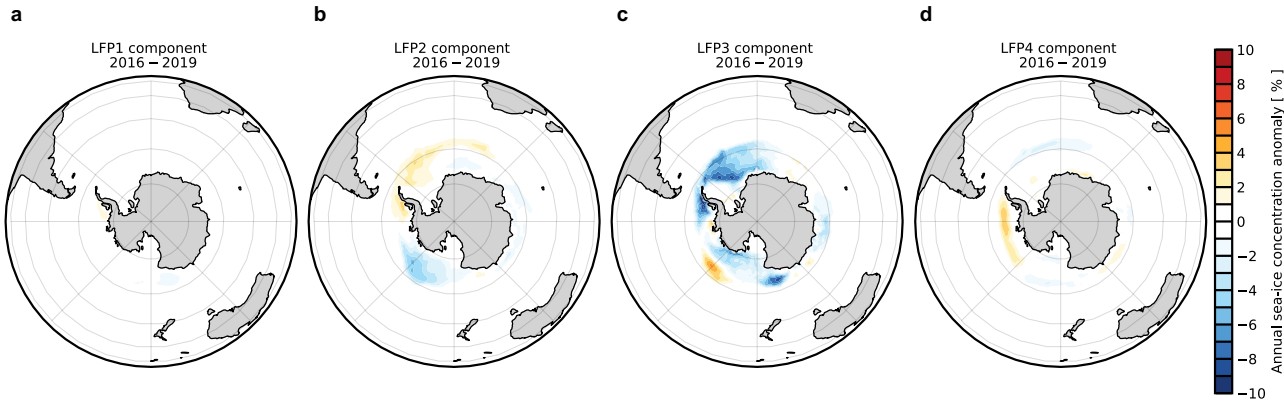

**Figure 4. Anomalies in low-frequency components of Antarctic sea-ice concentration.** Anomalies in annual-mean Antarctic sea-ice concentration during 2016 – 2019 relative to 1981 – 2010 associated with (a) LFC1, (b) LFC2, (c) LFC3, and (d) LFC4.

For pan-Antarctic SIA, LFC1 captures the long-term positive trend in Antarctic SIA (Fig. 5a) and accounts for approximately 20% of the total SIA variability (Fig. 5b). The dominant contributor to pan-Antarctic SIA variability, however, is LFC3 which captures the abrupt decline in SIA around 2016 (Fig. 5a) and accounts for approximately 60% of the total SIA variability (Fig. 5b). LFC3 also captures other periods of abrupt Antarctic SIA decline, such as the period around 1988 and closely follows the interannual variability of observed pan-Antarctic SIA. LFC2 contributes some to pan-Antarctic SIA variability (Fig. 5a),






accounting for approximately 5% of the total SIA variability (Fig. 5b). LFC4 contributes little to the pan-Antarctic SIA vari-
ability (Fig. 5a and 5b). Note both here and in the following paragraphs that the contributions from each LFC separately sums

to be slightly greater than the contributions of all four LFCs combined due to the non-orthogonality of the associated LFPs.

At regional scales, the influence of each LFC on Antarctic SIA variability is more distinct. In the Weddell Sea, for instance, all
four LFCs contribute to the SIA variability (Fig. 5c and 5h). LFC1 accounts for the gradual increase in SIA in the Weddell Sea
(Fig. 5c), while LFC2 accounts for higher-frequency variability, including the decline in the late 1990s. Both LFC1 and LFC2

account for approximately 10% of the SIA variability in the Weddell Sea. LFC3 again captures SIA anomalies associated with
the abrupt decline in 2016, and accounts for approximately 50% of the observed Weddell SIA variability. In the Indian and
West Pacific sectors each LFC accounts for much less of the SIA variability (Fig. 5d and 5e). The LFCA method also has
trouble reconstructing the observed SIA variability (see grey dotted lines in Fig. 5d and 5e), suggesting that variability in these
regions is contained in higher order EOFs. However, LFC1 does account for approximately 20% of the SIA variability in these

regions, which is likely related to the gradual positive trend in both timeseries (Fig. 5h). In the Ross Sea, the regional SIA
is reconstructed well (Fig. 5f). Here, LFC1 and LFC2 dominate the SIA variability capturing both the gradual positive trend
and interannual variability in SIA. In fact, LFC1 and LFC2 account for approximately 20% and 60% of the SIA variability,
respectively (Fig. 5h). Finally, in the Amundsen and Bellingshausen Seas, LFC1, LFC3, and LFC4 account for most of the
SIA variability (Fig. 5g and 5h). LFC1 captures the long-term decline in SIA and accounts for approximately 20% of the SIA

variability, while LFC4 captures the higher-frequency variability and accounts for approximately 60% of the SIA variability.

In summary, LFC1 accounts for the large-scale, gradual expansion of Antarctic sea ice and approximately all of the observed
long-term trends in Antarctic sea-ice concentration. The following three modes represent sea ice variability at progressively
shorter timescales. LFC2 accounts for most of the sea ice variability in the Ross Sea and contributes some to sea ice variability

in the Weddell Sea. LFC3 accounts for the abrupt decline in sea ice concentration around 2016 and captures most of the SIA
variability in the Weddell Sea and for pan-Antarctic SIA in general. Finally, LFC4 primarily captures sea ice variability in the
Amundsen and Bellingshausen Seas.

## 4  Mechanisms for low-frequency Antarctic sea-ice variability

### 4.1  Large-scale modes of climate variability

The distinct modes of variability in Antarctic sea-ice concentration associated with each LFC (Fig. 2) suggest distinct physical
mechanisms. To understand the processes related to each LFC, we regress annual-mean SST, annual-mean 500 hPa GPH, and
annual-mean near-surface winds onto each LFC at zero lag (Figure 6). We focus on the SST, 500 hPa GPH, and near-surface
wind fields first to understand the large-scale patterns of climate variability associated with each LFC. In Section 4.2 we ex-
amine more specific mechanisms related to changes in surface heat fluxes, ocean heat transport, and ocean mixed layer depth.






**Figure 5.** See next page.





**Figure 5. Low-frequency components of Antarctic sea-ice area.** (a) Annual-mean Antarctic sea-ice area anomalies computed from the first (blue), second (red), and third (yellow) LFPs and LFCs. The sum of the three components is also shown (grey dashed). (b) The proportion of variance explained in annual-mean Antarctic sea-ice area anomalies for the LFP1 (blue), LFP2 (red), LFP3 (yellow), and LFP4 (green) components. Regional Antarctic sea-ice area anomalies for five Antarctic regions: (c) Weddell, (d) Indian, (e) West Pacific, (f) Ross, and (g) Amundsen and Bellingshausen. The red subset denotes the geographical boundaries. (h) The proportion of variance explained in regional annual-mean Antarctic sea-ice area anomalies for the LFP1 (blue), LFP2 (red), LFP3 (yellow), and LFP4 (green) components. All panels use a 10-year cutoff and retain the five leading EOFs from 1979 to 2022. The thin bars in (b) and (h) denote the variance explained after removing a linear trend while the thick bars denote the variance explained for the full timeseries.

Each LFC exhibits a distinct spatial pattern of SST and 500 hPa GPH that is largely associated with Pacific and Southern Ocean climate variability (Figure 6). The SST pattern associated with LFC1 is reminiscent of the Interdecadal Pacific Oscillation (IPO), with a tripole-like SST pattern throughout the entire Pacific basin (Fig. 6a, left). This result supports Meehl et al. (2016) who argued that the phase of the IPO is a key source of the long-term positive trend in Antarctic sea-ice concentration from $2000-2014$. However, this SST pattern also resembles the observed monotonic SST trend over this time period, which climate models struggle to reproduce (Wills et al., 2022), raising the possibility that LFC1 could represent a transient response to anthropogenic forcing. The pattern of 500 hPa GPH associated with LFC1 exhibits strong positive values in the extratropics and negative values over the Southern Ocean (Fig. 6a, right), reminiscent of a Rossby wave train emanating from the tropical Pacific. In the Southern Ocean, LFC1 is associated with large-scale circumpolar surface cooling and negative 500 hPa GPH values indicating a strengthening of the circumpolar westerlies since the late 1970s (Fig. 6a, right). This is consistent with previous studies which have argued that surface cooling is linked to strengthening westerlies through enhanced northward Ekman heat transport (e.g., Hall and Visbeck, 2002).

The other LFCs exhibit various patterns of tropical Pacific variability, with little influence from the Atlantic basin (Fig. 6b-d, left). LFC2, for instance, has an SST pattern reminiscent of the central Pacific ENSO with negative SST anomalies centered in the equatorial Pacific basin (Fig. 6b, left). The SST pattern associated with LFC2 also has structure throughout the Pacific basin and somewhat resembles the North Pacfic Gyre Oscillation (NPGO), which is known to influence and be influenced by the central Pacific ENSO (Di Lorenzo et al., 2010). However, it is important to note that LFC2 is not purely the central Pacific ENSO and contains higher frequency variability associated with atmospheric circulation in the Southern Ocean. For instance, LFC2 has strong negative 500 hPa GPH anomalies in the Ross Sea which promotes surface cooling (Fig. 6b, right). This pattern of atmospheric circulation closely resembles the pattern of atmospheric circulation associated with the SAM (Fogt and Marshall, 2020), with negative 500 hPa GPH anomalies throughout the polar cap. This is also evident in LFC2, which exhibits strong positive values in the late 1990s, consistent with well-documented strong SAM and central Pacific ENSO events (e.g., Marshall, 2003; Fogt and Marshall, 2020). The SST and 500 hPa GPH pattern of LFC2 is consistent with well-known short-term Southern Ocean Ekman dynamics and the SAM, whereby stronger winds lead to surface cooling through enhanced





**Figure 6. Mechanisms for low-frequency variability in Antarctic sea-ice concentration.** Regression of annual-mean sea-surface temperature (color shading) and 500 hPa geopotential height field (green lines) onto the (a) first, (b) second, (c) third, and (d) fourth LFCs (10-year cutoff, 5 EOFs retained) in annual-mean Southern Hemisphere sea ice concentration from 1979 to 2022. The spacing for the 500 hPa geopotential height anomaly field is from -200 meters to 200 meters at 20 meter intervals. The left column shows the global domain and the right column shows the Southern Ocean domain. The red vectors on the panels in the right column denote the regression of annual-mean near-surface wind anomalies onto each LFC.



northward Ekman heat transport (e.g., Hall and Visbeck, 2002; Kostov et al., 2017).

The SST pattern associated with LFC3 is similar to the eastern Pacific ENSO with an elongated SST structure centered on the equator that extends across the Pacific basin (Fig. 6c, left). A pattern of positive SST anomalies in the Pacific (i.e., El Niño)
favors strong cooling throughout much of the Southern Ocean (Fig. 6c, right). Here, a positive phase of LFC3 is associated with weaker winds in the Ross Sea and stronger winds in the Weddell and Scotia Sea, and stronger meridional flow in the Amundsen and Bellingshausen Seas. The structure of the surface winds favors strong meridional heat and moisture transport in the Ross Sea and eastern edge of the Bellingshausen Sea. However, these surface wind patterns would also cause strong patterns of Ekman pumping and suction that affect the vertical transfer of heat. Note that negative SST anomalies in the Pacific
(i.e., La Niña) indicate strong surface warming in the Weddell and Scotia Seas and weak surface warming in the Indian and West Pacific sectors (Fig. 6c).

Finally, the SST pattern associated with LFC4 also exhibits a weak signature in the tropical Pacific, with a narrow band of negative SST anomalies on the equator in the central part of the Pacific basin (Fig. 6d, left). Interestingly, this SST pattern
coincides with the ENSO region, but does not exhibit the same degree of elongated SST structure as seen with LFC3 (compare Fig. 6c with 6d, left). LFC4 exhibits positive SST anomalies in the Amundsen and Bellingshausen Seas, promoting sea ice melt (Fig. 6d, right), with weaker SST changes in other regions of the Antarctic. The SST and GPH height anomalies associated with LFC5 (the last LFC) is shown in Figure A2. LFC5 exhibits a similar atmospheric circulation structure to the zonal wave three index with strong meridional flow in the Ross and Weddell Sea. LFC5 also exhibits little connection to global SSTs,
except for small cooling in the central Pacific. This suggests LFC5 represents much higher frequency atmospheric variability unique to the Southern Ocean.

In the next subsection, we re-examine how these modes of sea ice variability and the associated patterns of climate variability relate to periods of abrupt sea ice decline.

**4.2 Context for periods of abrupt decline**

In 2016, Antarctic sea-ice concentration experienced an abrupt decline (Fig. 1a and 1c). While this decline in Antarctic SIA clearly stands out in the satellite record, there have been other periods of abrupt sea ice loss, such as in the late 1980s and throughout the mid- to late- 2000s. In Section 3, we showed that LFC3 accounts for most of the abrupt SIA changes (Fig. 5a and 5b). We now re-examine sea ice concentration anomalies associated with each LFC, but focus on changes over periods
of abrupt decline. These periods were identified as 4-year time periods where LFC3 transitioned from above one standard deviation to below one standard deviation. This resulted in two low-to-high ($L - H$) composites: 1989/1990 minus 1987/1988 and 2017/2018 minus 2015/2016. In the subsection below we refer to each as Event 1 and Event 2, respectively.



Figure 7 shows the Antarctic sea-ice concentration change for these two time periods in observations (Fig. 7a) and in each
LFC (Fig. 7b-e). During Event 1, negative sea ice concentration anomalies occurred throughout much of the Antarctic (Fig.
7a, left). However, positive sea ice concentration anomalies also occurred in the Weddell Sea, Indian sector and parts of the
Amundensen and Bellingshausen Seas (Fig. 7a, left). During Event 2, on the other hand, there was a similar negative sea ice
concentration anomaly pattern, but no corresponding positive sea ice concentration anomalies in the Weddell Sea and parts of
the Amundensen and Bellingshausen Seas (Fig. 7a, right). The same $L - H$ composites for each LFC show that these slightly
different sea ice concentration anomaly patterns, and therefore different SIA declines, arise from LFC1. During Event 1, LFC1
contributes large positive sea ice concentration anomalies throughout much of the Antarctic, while during Event 2, LFC1 did
not contribute to change in sea ice concentration. Both LFC2 and LFC3 exhibit similar magnitudes of sea ice concentration
change for Event 1 and Event 2, showing negative sea ice concentration anomalies in the Ross Sea from LFC2 (Fig. 7c) and
throughout much of the Antarctic from LFC3 (Fig. 7d). This suggests that one reason Event 1 was not as anomalous as the
2016 event was because of counteracting modes of tropical variability (LFC1 and LFC3), which prevented large negative sea
ice concentration anomalies from emerging in the Weddell Sea and Amundsen and Bellingshausen Seas.

To better understand the processes responsible for the different sea ice concentration anomalies for these two time periods,
we examine components of a Southern Ocean mixed-layer temperature $T_m$ budget (assumed to be equal to the SST). We
assume sea ice concentration anomalies are related to SST anomalies. Following Pellichero et al. (2017), the rate of change of
$T_m$ can be expressed in terms of air-sea fluxes, horizontal advective fluxes (geostrophic, ageostrophic, and Ekman), vertical
entrainment at the base of the mixed layer $h_m$, and diffusive processes:

$$\frac{\partial T_m}{\partial t} \approx \frac{Q_s}{\rho_0 c_p h_m} - \mathbf{u}_m \cdot \nabla T_m - w_e \frac{\Delta T}{h_m} + \kappa \frac{\partial^2 T_m}{\partial z^2}, \tag{1}$$

where $Q_s$ is the net surface heat flux into the mixed layer, $\rho_0$ is a reference density of seawater, $c_p$ is the specific heat of seawater,
$\mathbf{u}_m$ is the mixed-layer averaged horizontal velocity (including geostrophic and Ekman components), $w_e$ is the entrainment
velocity associated with a variable mixed layer depth, $\Delta T$ corresponds to the temperature differences across the base of the
mixed layer, and $\kappa$ is the vertical turbulent diffusion coefficient at the base of the mixed layer. The entrainment velocity $w_e$ can
be calculated from the rate of change in $h_m$ following Ren and Riser (2009):

$$w_e = \frac{\partial h_m}{\partial t}. \tag{2}$$

It is possible to explicitly calculate each term in Eq. (1) and assess their contribution to temperature changes in Events 1 and 2.
However, this is difficult with existing data products, particularly for early parts of the satellite record, and it is not the primary
focus of this study. As noted by Tamsitt et al. (2016) and Pellichero et al. (2017) large uncertainties arise for a number of terms
in Eq. (1) when using reanalysis products, making it difficult to close the temperature budget without taking a zonal average.



Furthermore, estimates of $\kappa$ and geostrophic velocities in the sea ice zone are weakly constrained and difficult to determine in observations. Instead, we plot the $L - H$ composites of $T_m$ as well as various physical properties, such as the net surface heat flux, zonal-wind stress, and ocean mixed layer depth, which contribute to components of Eq. (1). We assume that Ekman transport is the dominant contributor to the advection term in Eq. (1), particularly at scales much larger than the ocean mesoscale. The key changes in these composite properties are discussed below. However, we acknowledge that there may be significant contributions to mixed-layer temperature changes that arise from other terms, such as the vertical temperature difference across the base of the mixed layer, the turbulent diffusivity, and geostrophic velocities.

Both abrupt decline events are associated with positive SST anomalies in the Scotia Sea (north of the Weddell Sea), but Event 2 exhibits more circumpolar warming when compared to Event 1, which exhibits negative SST anomalies in the Ross Sea and West Pacific and Indian Sectors (Fig. 8a). Differences in Southern Ocean circumpolar westerlies likely explain the different characteristics of these two abrupt sea ice decline periods (Fig. 8). In the Southern Ocean, for Event 1, the SST anomalies are somewhat correlated with surface heat flux anomalies (Fig. 8b, left), while for Event 2, the SST anomalies are less correlated with surface heat flux anomalies (Fig. 8b, right). In fact, Event 2 is mostly related to weakened circumpolar westerlies in the Southern Ocean (Fig. 8c, right), which would cause an anomalous convergence of heat along the margins of the sea ice edge due to anomalous southward Ekman transport. Furthermore, the ocean mixed layer depth shoaled more broadly and to a greater extent in Event 2, as compared to the mixed-layer depth change that occurred during Event 1 (Fig. 8d). This larger shoaling would amplify the warming associated with surface heat flux and Ekman changes, and it would cause mixed layer warming through reduced entertainment of cold waters, as noted in Eq. (1). The anomalous shoaling of the ocean mixed layer is consistent with a reduction in westerly wind strength (Wilson et al., 2023).

Indeed, the different changes between Event 1 and Event 2 can be inferred from Figure 2 and Figure 6. When LFC1 is in a positive phase, meaning there are strong positive sea ice concentration anomalies throughout much of the Antarctic (Fig. 2a), near-surface winds strengthen and there is surface cooling (Fig. 6a, right). This cooling counteracts warming due the weakening of the near-surface winds that is associated with LFC3 (Fig. 6c, right). Figure 8 shows that Event 1 — which is a combination of LFC1 and LFC3 — exhibits little-to-no weakening of the circumpolar westerlies, while Event 2 — which is mostly LFC3 — exhibits strong weakening of the circumpolar westerlies. The counteracting of LFC1 and LFC3 in Event 1 prevents both reduced northward Ekman heat transport and a shoaling of the ocean mixed layer, which supports near-circumpolar warming throughout the Southern Ocean and likely circumpolar sea ice loss. Additional work is required to understand how other mechanisms, such as geostrophic transport and eddy mixing, impact the pattern of SST anomalies for these two time periods.

## 5 Discussion and conclusions

The recent decline in Antarctic sea ice, which occurred after a gradual, long-term increase, demonstrates that, under certain conditions, Antarctic sea ice can be susceptible to rapid changes. It is well understood that Antarctic sea ice exhibits substan-





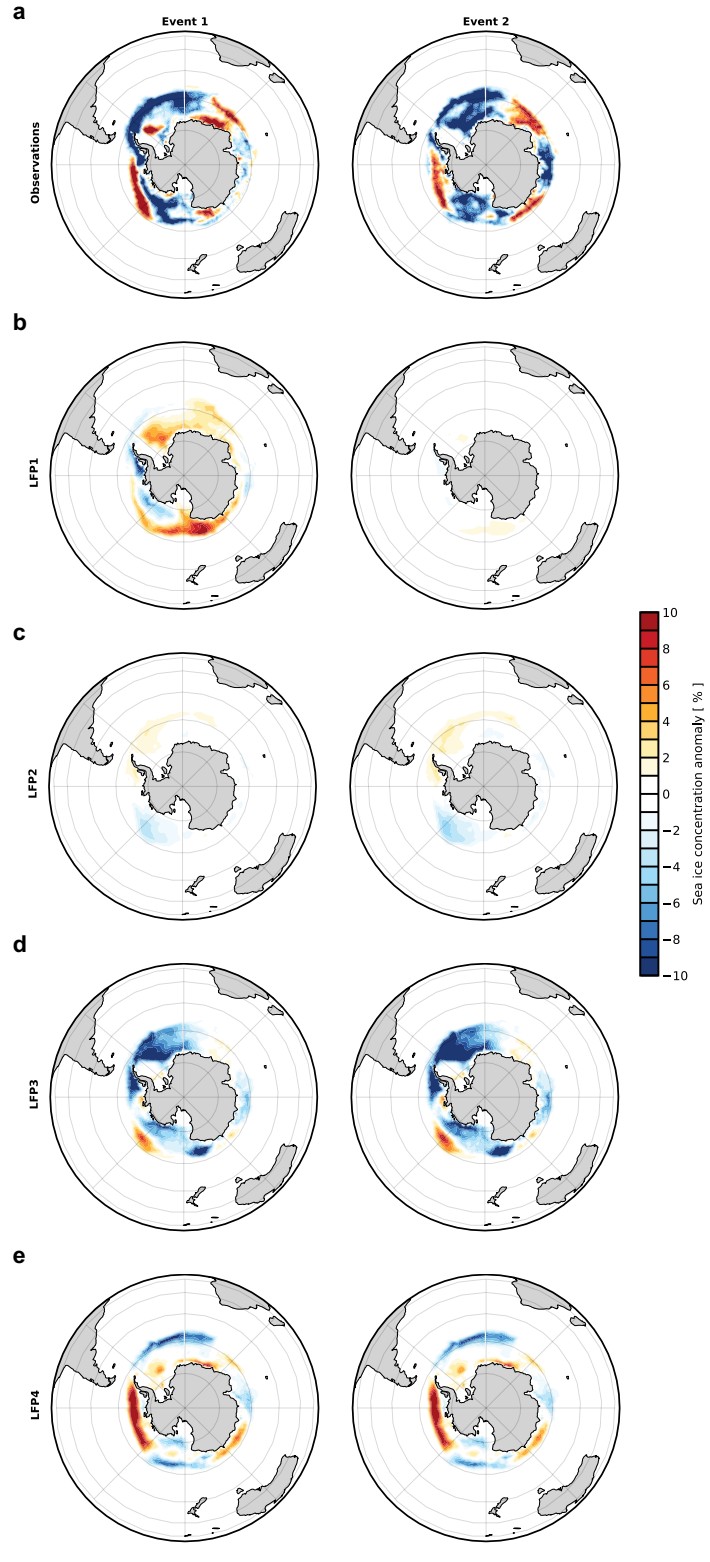

**Figure 7.** See next page.



**Figure 7. Periods of abrupt Antarctic sea-ice decline.** Changes in annual-mean Antarctic sea-ice concentration from (a) observations and from each (b-e) LFP component for (left) $1987 - 1990$ and (right) $2015 - 2018$. Changes are calculated as differences between the first two years and last two years of each period. Event 1 is 1989/1990 minus 1987/1988 and Event 2 is 2017/2018 minus 2015/2016. These periods were identified as 4-year time periods where LFC3 transitioned from above one standard deviation to below one standard deviation.

tial interannual-to-decadal variability, but the precise mechanisms responsible for this variability and how these mechanisms
manifest spatially and temporally in the observational record has remained unclear. In this paper, we used a novel statistical method (i.e., LFCA; Wills et al., 2018; Schneider and Held, 2001) to identify patterns of low-frequency variability in observed Antarctic sea-ice concentration. We identified distinct modes of low-frequency Antarctic sea-ice variability. The leading mode represents the large-scale, gradual expansion of Antarctic sea-ice. This mode accounts for approximately all of the observed trends in Antarctic sea-ice concentration and long-term trends in regional and total Antarctic SIA. The next three modes rep-
resent higher-frequency sea ice variability. The second mode (LFC2) accounts for most of the sea ice variability in the Ross Sea and contributes some to sea ice variability in the Weddell Sea. The third mode (LFC3) accounts for the abrupt decline in sea ice concentration around 2016 and captures most of the SIA variability in the Weddell Sea and for pan-Antarctic SIA. The fourth mode (LFC4) accounts for sea ice variability mainly in the Amundsen and Bellingshausen Seas.

We then identified large-scale atmospheric and oceanic mechanisms associated with each mode, including process that related to the gradual expansion and abrupt decline of Antarctic sea-ice concentration. All LFCs are influenced to some degree by tropical Pacific variability, with little influence from the Atlantic basin. The SST pattern associated with LFC1 is reminiscent of the IPO, featuring a tripole-like SST pattern across the Pacific basin. This is consistent with Meehl et al. (2016) which argued that observed Antarctic sea-ice expansion is related to the IPO phase. The spatial pattern of SST associated with LFC1
also resembles the observed trend in SST that climate models struggle to reproduce (e.g., Wills et al., 2022), meaning biases in Antarctic sea ice and global SST trends are likely related. However, it is still unclear whether global SST trends are the cause of or the result of Southern Ocean trends. Dong et al. (2022a) showed that Southern Ocean cooling can cause a two-way teleconnection that results in a similar SST pattern to observed trends. This implies that other sources of Southern ocean cooling, such as increased surface freshening (Pauling et al., 2016; Haumann et al., 2020; Dong et al., 2022b), might also impact
global SST trends. Nonetheless, the anomalous circulation associated with LFC1 indicates a strengthening of the circumpolar westerlies and favors strong surface cooling throughout much of the Southern Ocean. This is also consistent with recent work that has argued near-surface wind trends are a key ingredient of observed Antarctic sea-ice expansion and Southern Ocean cooling (e.g., Blanchard-Wrigglesworth et al., 2021; Sun and Eisenman, 2021).

The other LFCs are related to higher-frequency Pacific variability. LFC2 is shown to be related to the central Pacific ENSO and SAM variability, with a strong atmospheric circulation pattern centered in the Ross Sea and Weddell Sea. LFC3 — the mode that accounts for the abrupt decline in sea ice around 2016 — has a SST pattern reminiscent of the eastern Pacific ENSO. This mode favors strong surface warming in and around the Weddell Sea when the eastern Pacific ENSO is in a negative phase, and

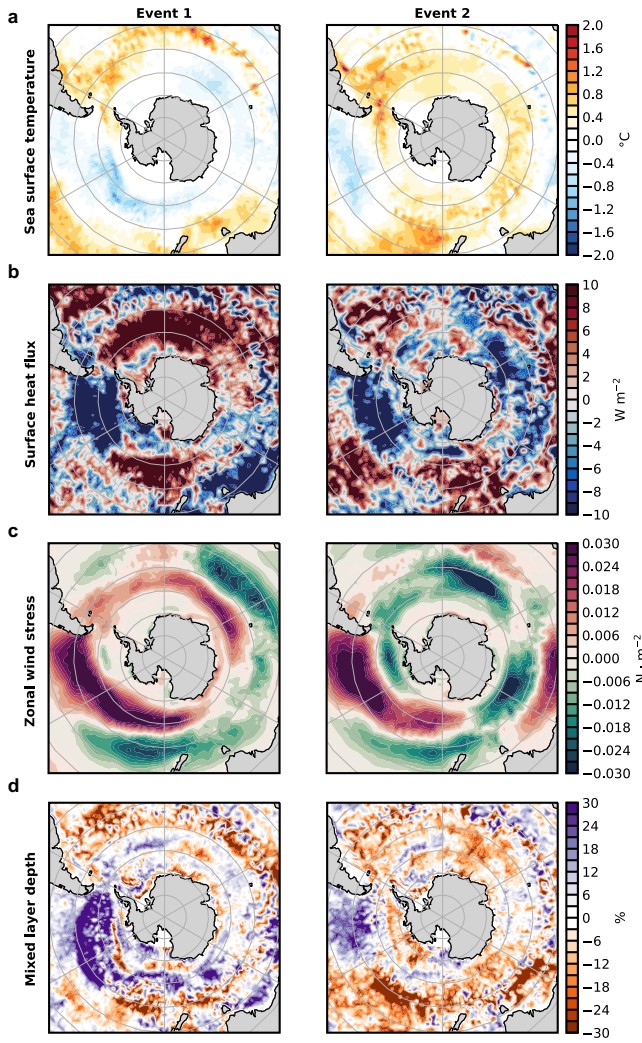

**Figure 8. Processes associated with periods of abrupt Antarctic sea-ice decline.** Changes in annual-mean (a) sea surface temperature, (b) net surface heat flux, (c) zonal wind stress, and (d) ocean mixed layer depth for (left) $1987 - 1990$ and (right) $2015 - 2018$. The ocean mixed layer depth anomalies are normalized by the climatology. Changes are calculated as differences between the last two years and first two years of each period. Event 1 is 1989/1990 minus 1987/1988 and Event 2 is 2017/2018 minus 2015/2016.





strong surface cooling when the eastern Pacific ENSO is in a positive phase. This mode also contains a signature of Amundsen
Sea Low variability with a strong pattern of atmospheric circulation near the Amundsen and Ross Seas (Raphael et al., 2016).
We showed that this mode accounts for the 2016 decline event through a weakening of circumpolar westerlies which favors
anomalous southward Ekman heat transport and shoaling of the ocean mixed layer depth, both of which cause warming and
sea ice melt. Finally, LFC4, which is more localized to the Amundsen and Bellingshausen Seas, is weakly related to other
tropical variability in the central equatorial Pacific. Interestingly, this mode of Pacific SST variability occurs in a similar region
that other studies have argued strongly impacts the Amundsen and Bellingshausen Seas (e.g., Steig et al., 2012; Dutrieux et al.,
2014; Holland et al., 2019). These studies have argued that increased ice-shelf melt in the Amundsen and Bellingshausen Seas
arises from increased poleward ocean heat transport driven by central equatorial Pacific climate variability. Our work suggests
that this mode also impacts Antarctic sea ice and is distinct from the central Pacific ENSO.

While these results demonstrate the utility of LFCA for interpreting interannual-to-decadal sea ice variability, in this paper we
focused only on annual-mean Antarctic sea ice. Antarctic sea ice exhibits strong seasonality, with larger positive sea ice trends
in the austral autumn and enhanced sea ice variability in the austral winter (Holland, 2014). Recent work has also shown that
the seasonality of near-surface winds and mixed-layer depth in conjunction with surface heating can produce large and abrupt
circumpolar surface warming in the Southern Ocean (Wilson et al., 2023), which could also cause abrupt sea ice changes. Other
studies have shown that seasonal changes in ENSO and other modes of climate variability can also exert strong control on sea
ice changes (e.g., Stuecker et al., 2017; Schlosser et al., 2018). In particular, Stuecker et al. (2017) argued that the persistence of
positive SST anomalies in the Ross, Amundsen and Bellingshausen Seas from a positive ENSO phase in December–February
contributed to the abrupt decline in Antarctic sea ice in 2016. Examining sources of low-frequency Antarctic sea-ice variability
on seasonal timescales might improve mechanistic interpretation of the observational record. Such work might also explain
why other periods with large ENSO events, such as 1997/1998, did not result in large Antarctic sea-ice changes. Our work
suggests that the flavor of ENSO can result in different Antarctic sea-ice concentration changes, and further examination on
seasonal timescales might also reveal unique sea ice changes for different ENSO events. However, interpreting sea ice patterns
from LFCA on seasonal timescales might be difficult, particularly in the austral winter when ocean reanalysis is poorly con-
strained.

Although these results do not provide a full mechanistic pathway to explain abrupt Antarctic sea-ice changes, the results do
provide context for periods of abrupt decline that might inform climate model experiments. Our results suggest that SST
variability in different regions of the Pacific can result in regionally-distinct Antarctic sea-ice concentration anomalies that
sometimes counteract each other. These results could inform so-called "pacemaker" experiments (e.g., Kosaka and Xie, 2013),
where SSTs are relaxed to observed values to isolate key regions of influence on climate variables. Performing these experi-
ments over various regional domains of the Pacific might demonstrate the importance of phasing in Pacific climate variability
in contributing to abrupt sea ice changes. Such experiments might also better elucidate the mechanisms underpinning Antarctic
sea-ice variability and help to clarify why certain periods, such as the late 1980s or 1990s, did not result in wide-spread and



abrupt sea ice loss. This work might also help to clarify whether observed Antarctic sea ice has experienced a regime change
or not (e.g., Raphael and Handcock, 2022; Fogt et al., 2022) as multiple simulations can be performed providing context for
internal variability.

The utility of the LCFA is that it can identify modes of variability related to both the gradual expansion and abrupt decline of
observed Antarctic sea-ice concentration. This method and framework thus opens up other avenues of sea ice research, such
as model bias diagnosis and ice-ocean-atmosphere interactions. For example, using this method in conjunction with climate
models might help identify mechanisms responsible for the large discrepancies in Antarctic sea ice trends between climate
models and observations over the satellite record (e,g., Purich et al., 2016; Rosenblum and Eisenman, 2017; Roach et al.,
2020). Our results show the SST pattern associated with LFC1 — which captures the long-term expansion of Antarctic sea ice
— resembles the SST trend bias in state-of-the-art climate models (see Fig. 1 in Wills et al., 2022), which suggests biases in
the SST and Antarctic sea ice trends of climate models are related to the same physical processes. Our results also highlight
mechanisms impacting short-term sea ice changes. The large influence of ENSO on abrupt sea ice changes suggests that as
ENSO transitions from its negative phase $(2020-2022)$ to its positive phase, Antarctic sea ice cover might increase, potentially
offsetting the anomalous decline seen since 2016. However, this analysis indicates that the impact of ENSO on Antarctic sea
ice depends on whether ENSO manifests more in the central or eastern Pacific, suggesting it will be crucial to monitor regional
Pacific SST variability for short-term predictions of Antarctic sea-ice changes.

*Code and data availability.* All data in this study are publicly available. Monthly Antarctic sea-ice concentration is available through the
National Snow and Ice Data Center (https://doi.org/10.7265/efmz-2t65). The ERA5 reanalysis data is available through the Copernicus
Climate Change Service (https://climate.copernicus.eu/). Code for LFCA is available on GitHub (https://github.com/rcjwills/lfca).

*Author contributions.* D.B.B., J.D., R.C.J.W, and M.Å conceived the study. D.B.B. performed the analysis, produced figures, and wrote the
paper. All authors contributed to the methods design, interpretation of the results, and manuscript reviewing.

*Competing interests.* The authors declare no conflict of interests.

*Acknowledgements.* D.B.B thanks the Nansen Legacy Project for funding part of this research through a visit to University of Bergen.
D.B.B. was supported by the National Science Foundation Graduate Research Fellowship Program (NSF Grant DGE-1745301). J.D. and
M.Å. were funded by the Research Council of Norway projects Nansen Legacy (Grant 276730) and the Trond Mohn Foundation (Grant
BFS2018TMT01). R.C.J.W. was supported by the National Science Foundation (NSF Grant AGS-2203543). A.F.T. was supported by the
Office of Naval Research's Multidisciplinary University Research Initiative (N00014-19-1-2421).





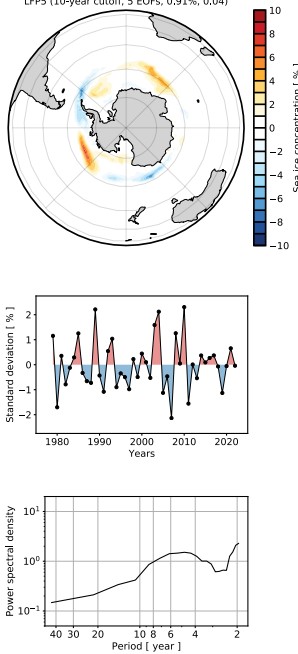

**Figure A1. The additional low-frequency component and pattern.** Fifth (top) low-frequency pattern (LFP), (middle) low-frequency component (LFC), and (bottom) power spectra density of the LFC using a 10-year cutoff and retaining the five leading EOFs with annual-mean Antarctic sea-ice concentration anomalies from 1979 to 2022. Power spectra are computed with multitaper spectral analysis (Percival et al., 1993). The fraction of explained low-frequency variance (in %) and the ratio $r$ of low-frequency to total variance is given for each pattern.



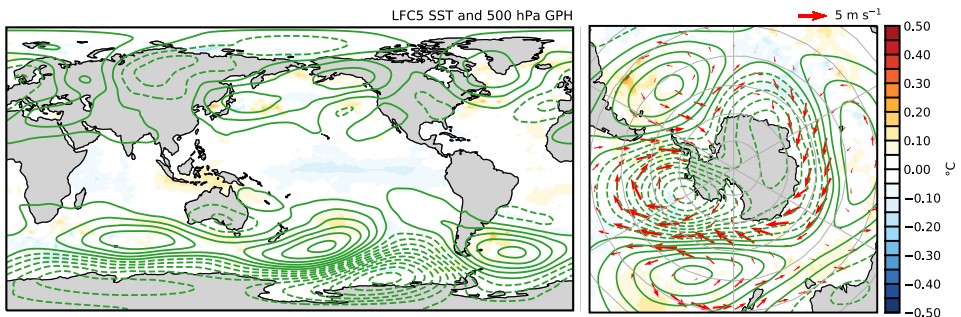

**Figure A2. Mechanisms for additional low-frequency variability in Antarctic sea-ice concentration.** Regression of annual-mean sea-surface temperature (color shading) and 500 hPa geopotential height field (green lines) onto the fifth LFC (10-year cutoff, 5 EOFs retained) in annual-mean Southern Hemisphere sea ice concentration from 1979 to 2022. The spacing for the 500 hPa geopotential height anomaly field is from -200 meters to 200 meters at 20 meter intervals. The left column shows the global domain and the right column shows the Southern Ocean domain. The red vectors on the panels in the right column denote the regression of annual-mean near-surface wind anomalies onto each LFC.

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
