# Peer review of "Sources of low-frequency variability in observed Antarctic sea ice"

_EGUsphere, 2023_

## Author Response (AR1)

**Sources of low-frequency variability in observed Antarctic sea ice**

**Response to Reviewers**

We thank the three reviewers and the Editor for carefully reading the manuscript and for their encouraging suggestions. Below, we respond directly to the particular comments of each reviewer, using black text to indicate text copied verbatim from the reviewer's report and blue text to indicate our responses. All line numbers referenced in this response refer to the tracked manuscript.

**Response to Reviewer 1**

This is an interesting manuscript that applies a novel methodology (LFCA) in the context of Antarctic sea-ice analysis. It provides results that confirm earlier ones, such as the influence by IPO and ENSO, and new results, such as the co-variability of LFC modes and identifies a mode that climate models struggle to reproduce, a finding that is particularly useful to climate modellers.

Thank you for the encouraging comments. Below we respond to your major and minor comments.

One potentially major issue is the robustness of the findings. The current analysis is based on one sea-ice concentration data from NSIDC, one atmospheric reanalysis (ERA5), and one ocean reanalysis (ORAS5). It is known that the Antarctic sea-ice concentration data has significant uncertainties (https://doi.org/10.1029/2011GL047553), as do reanalyses (https://doi.org/10.1007/s00382-018-4242-z). It would be important to estimate how much these uncertainties affect the findings of this study.

Thank you for bringing up this point. We conducted the same analysis using OSI SAF sea ice concentration and the LFC and LFPs are very similar. We also regressed the new LFCs from OSI SAF onto other reanalysis products to examine the large-scale modes of SST variability. These modes are also similar. We have added a few lines to the manuscript to acknowledge this (see Lines 80-85).

Other than the aforementioned issue, some minor comments and suggestions could be considered as listed next in no particular order if importance:

- line 30. The abrupt decline still persists as February 2023 SIA was record low.
  We have altered this sentence to reflect the record low in the year 2022 (see Lines 30-35).

- line 73. As LFCA is based on PCA it is a statistical method and can not determine physical and dynamical processes causing sea-ice variability.
  We have refined the language to acknowledge LFCA is still statistical (see Lines 70-75).

- lines 116-121. Could be further clarified how the explained SIC variance by LFPs compare with explained variance of LFP SIA time series. For example, why LFP1 has the highest variance ratio, while LFP3/LFC3 explain the highest proportion of pan-Arctic SIA variance.
  We have added a statement to the manuscript that clarifies here that r is the ratio of low-frequency to total variance and thus does not guarantee that the first LFC accounts for the most sea ice variability, which could be high frequency (see Lines 115-120).

- line 135. Did you try other than 10-year cutoff filter and 5 leading EOFs?
  We did examine how various combinations of EOFs and cutoffs affect the results. This is acknowledged and described further in Lines 115-120.

- line 149. LFC3 during the abrupt sea-ice decline coincides with low LFC1 values which could perhaps mentioned.
  We have acknowledged the similarities between each LFC at the abrupt decline period (see Lines 150-155).

- line 189. '... contributions from each LFC to SIA separately ...'
  We fixed this (see Lines 195).

- line 200. Some of these trends look rather small in Figure 5, e.g. in 5h. Are they statistically significant and therefore worth of discussing?
  Figure 5 and Figure 5h in particular show the variance each LFC-based SIA accounts for in the total SIA timeseries. It is not showing the trend. We have refined the text to make sure this is clear (see Lines 200-210).

- In some figures, SIC anomaly colourbars are from blue to red (e.g. Fig. 2), in some others red to blue (e.g. Fig 1.). Would be reader friendly to have similar colourbars in all SIC anomaly plots.

  Thanks for catching this. We changed Figure 1 to have the same colorbar as all sea ice concentration anomaly plots.

- line 251. Stronger winds in the Weddell and Scotia Seas are not apparent in Fig. 6c.

  We have revised the language in this section by removing the word stronger and using anomalies instead (see Lines 245-255).

- lines 253-254. Also LFC2 wind patterns would cause strong Ekman pumping and suctions but that is not discussed.

  We have added a statement about this (see Lines 245-255).

- line 255. Would be useful to discuss how LFC3 effects are transmitted from the Pacific sector to the Weddell and Scotia Seas.

  We do not understand this comment, but interpret it as inquiring about the atmospheric "teleconnection"'. We have edited the manuscript to better describe what LFC3 means (see Lines 255-300).

- line 265. 'much higher' seems exaggeration, 'higher' is better.

  Thanks. We changed this.

- line 266. '... the Southern Ocean than LFC1-4.'

  We have rephrased this sentence.

- line 284. The event2 positive sea-ice concentration anomaly in the Amundsen Bellingshausen Seas is almost as strong and somewhat more extensive than during event1.

  Thanks for noting this. We describe this later on in the section (see Lines 290-300).

- line 284. 'Amundensen' to 'Amundsen'.

  Fixed.

- line 300. Would be good to mention that the positive flux is directed downward.

  Done.

- line 315. '... geostrophic velocities and mesoscale eddies.'

  We have changed this.

- line 326. Term 'anomalous southward Ekman transport' is misleading because the Ekman transport is northward directed. 'Weaker northward Ekman transport' would be better.

  We have changed this.

- line 345. A novel statistical method is not 22 year old. It is novel in the context of Antarctic sea-ice analysis.

  We removed the work novel.

- line 346. '...patterns and distinct modes of...'

  Done.

- line 347. Delete sentence 'We identified distinct modes of low-frequency Antarctic sea-ice variability'.

  Done.

- line 363. Change 'Southern ocean' to 'Southern Ocean'.

  Thanks. We corrected this.

- line 374. Fig. 6c shows rather modest looking surface warming, while Figure 2c shows remarkable sea-ice loss. Note that sea-ice loss may not necessarily be associated with strong SST increases as polar ocean surface temperatures tend to stay close to the freezing point.
  Thanks for pointing this out. We have changed the wording in this section to reflect that SST and SIC anomalies are not necessarily the same (see Lines 345-350).

- line 398. Mention here that ORAS5 was used.
  Done.

**Response to Reviewer 2**

This paper describes an analysis using a novel method involving low-frequency component analysis to determine spatial-temporal structures of Antarctic sea ice variability. The authors aspire to get to the causes of the gradual increase of Antarctic sea ice extent during the satellite era from 1979-2016, and then the abrupt decline that started in 2016. They quantify a number of aspects of Antarctic sea ice variability in terms of spatial evolution and timescales involved, and use that information to attempt to connect to possible processes responsible.

Major comment: The starting point for this analysis is a statistical method, and then the authors proceed to attempt to come up with physical processes that could explain the results of the statistical analysis. This is more or less the reverse of what would usually be done in a scientific study. That is, first a physical hypothesis should be posed, and then statistical methods are chosen to determine whether the hypothesis can be supported or not. The latter can involve the use of the statistical methods to point to supporting physical mechanisms that can substantiate (or not) the hypothesis. Since the authors start with the statistical analysis, and since there is no hypothesis posed to frame the analysis, it's unclear where they are headed with the analysis. In the end, they backfill some physical mechanisms to try to explain their statistical results. I don't think the authors have to re-do any of their analyses, but they just need to frame and organize their results in a better way. First, what is the scientific hypothesis they want to test, for which they employ their statistical methods? Then the methodological results can be described, after which those results could point to some physical processes involved with the hypothesis they are testing. In the end, they can conclude with what they have learned about the processes involved with the hypothesis, and how this advances our knowledge about Antarctic sea ice.

Thank you for bringing up this point about hypothesis driven research and lack of a specific research question. In the revised draft we have improved the introduction to emphasize the significant variability that Antarctic sea ice has experienced over the satellite record and how it motivates this work to understand the sources of this variability (see Lines 25-32). We have also improved discussion of the mechanisms that have been proposed in controlling Antarctic sea-ice variability and explain how it is unclear how these modes relate to the spatial-temporal variability (see Lines 35-70). We have also improved the description of the statistical method (e.g., LFCA) and why it is different from other work (Lines 70-78).

Detailed comments:

- Figure 2 caption: at the end of the caption, the authors should add, "…in the titles of the top panels" to be clear where those numbers appear that they are referring to
  We have added this to the caption of Figure 2 and Figure A1.

- Line 272: But the massive decline of 2016 eclipses other "abrupt" declines since 1979, so is it fair to compare other periods to 2016? Perhaps a few caveats would be in order here.
  Thank you for bringing up this point. This is exactly why we are re-examining past periods of abrupt decline. While the 2016 period was the largest on record, when using only LFC3, it can be argued that the late 1980s also has a big event. We use this as motivation to investigate why the total (i.e., the sum of all LFCs) was not the same and show that LFC1 modulated the magnitude of the event (see Lines 278-350). We have refined parts of this subsection to better emphasize why we are comparing these two time periods (see Lines 278-282, 340-350).

- Line 295: In many of the references cited by the authors, the role of meridional winds in contributing to variations of Antarctic sea ice extent is described. For example, if there are north winds (in the eastern Amundson Sea with a strong ASL), there is warm air advection as well as mechanical wind forcing to make the sea ice retreat. This is related to the strength of the westerlies of course because if there are strong meridional wind anomalies, the mean

westerlies are weakened. But the role of meridional winds should at least be mentioned here since they play such a prominent role in the literature regarding regional advances and retreats of Antarctic sea ice.

Thank you for bringing up this point. We have changed parts of Section 4.1 and 4.2 to acknowledge the role of meridional winds rather than purely zonal winds and Ekman dynamics (see Lines 259-261, 270-271 and Lines 345-355). If there are specific papers regarding the role of meridional versus zonal winds that the reviewer is aware of, we would appreciate being pointed to the specific literature they are referring to.

**Response to Reviewer 3**

This manuscript considers the observed timeseries of Antarctic sea ice and its decomposition into low-frequency modes of variability via low-frequency component analysis, LFCA. This statistical method identifies modes of variability based on their dominant timescale, and has not previously been applied to Antarctic sea ice. It yields some insights into the post-2016 decline relative to another period of decline in the late 1980s. The major mode accounts for the large-scale sea ice expansion and is associated with IPO-like SSTs and strengthening westerlies. Other modes are associated with SAM and ENSO.

The manuscript is interesting, clear and well-written, and a good fit for The Cryosphere. Many of the results support previous studies, rather than being particularly novel; however, given the large uncertainty in observed Antarctic sea ice evolution, this is still is a welcome contribution. It would be interesting to see this analysis applied to climate models in future work.

*Thank you for the encouraging comments. We agree it would be interesting to apply this to climate models in future work. Below, we respond to your major and minor comments.*

Major comments

- The authors should give some indication of how the results may be affected by observational uncertainty. This should be quick to do with the NSDIC sea ice concentration — that release contains results from the NASA Team and Bootstrap algorithms, which could be used to show any differences from the Climate Data Record. Similarly, it would be good to know how the ORAS5 reanalysis compares to ARGO mixed layer depths. They should also briefly mention potential biases in the ERA5 reanalysis.

  *Thanks for bringing up this. Reviewer 1 also brought up concerns about observational uncertainty in sea ice concentration datasets. We conducted LFCA with another sea ice concentration product (OSI-SAF). There results are very similar and now acknowledged in the methods section (see Lines 80-85). We also added a few statements about the mixed-layer heat budget analysis at the end of manuscript that pertains to the potential biases in mixed layer depth (see Lines 324-326).*

- The authors suggest an important role for ENSO in the post-2016 decline. However, Wilson et al. (2023, cited in manuscript) emphasizes the role of the SAM more so than ENSO. I would like to see more discussion on the findings of Wilson et al. 2023 in relation to this work. I note the two studies share some co-authors.

  *Thank you for noting this. We emphasize that Wilson et al., (2023) based their analysis on the sea ice free zone of the Southern Ocean, while our analysis pertains almost entirely to sea ice. Thus, a comparison of these two studies is not apples-to-aples. However we agree that the two concusions are interesting and warrent more discussion. To this end, we have changed the Discussion section to better reflect the relative role of ENSO/SAM and seasonality as discussed in Wilson et al., (2023) (see Lines 400-408).*

- I thought the comparison of Event 1 and Event 2 in Sec. 4.2 was particularly interesting. However, this didn't make it into the Discussion and Conclusions. I would suggest adding it. The authors may want to consider separating the Conclusions from the Discussion, which may help to more clearly state their contributions.

  *Thanks for pointing this out. We meant to emphasize this result. We have added a statement to the discussion to acknowledge this (see Lines 386-391).*

Minor comments:

- L10: Change 'This mode' to 'This third mode' for clarity
  Done.

- L13: 'global sea-surface temperature' - In several places in the manuscript, the authors mention global SST temperatures, implying there are systematic model biases in global SST trends. Is there a reference for this (as opposed to Southern Ocean SST trends)? There is no analysis using global SSTs in the manuscript.
  We apologize for the confusion. We used the word "global" to refer to SST trends everywhere or the large-scale SST trends. We have revised the manuscript to acknowledge it is mainly the Southern Ocean and when applicable we change "global" to "large-scale" to emphasize that we mean SST trends everywhere.

- L79: Which NSIDC dataset is used? The Data Availability section suggests version 4. This should be noted in the manuscript, and in this case the citation should be Meier et al. 2021 (see the NSIDC download page). If it is version 3, the authors should update to version 4.
  Thanks for catching this. We are using the latest version. We have altered the manuscript to acknowledge this (see Lines L81-83).

- L86: The authors might preview that the choice of annual mean over seasonal analysis will be discussed further at the end, as this was an initial concern when reading the manuscript.
  We have now noted this in the methods section (see Lines 91-92).

- L115: 'mixes modes of variability that appear to be distinct' - how is this judged
  We have changed this sentence to emphasize that reducing the number of EOFs mixes modes of ENSO variability that appear to be distinct (Lines 115-120).

- L225: 'this SST pattern also represents the observed monotonic SST trend over this time period' - which SST trend is referred to here? The Southern Ocean average SST trend is not monotonic
  We have removed the word 'monotonic' and also rephrased the statement to acknowledge this is about global SST trends.

- L239: 'circumpolar' - this pattern is certainly stronger in the Pacific than other regions
  We have rephrased this sentence to emphasize it is more local to the Pacific.

- L230 and L245 - these sentences on the impact of winds are somewhat repetitive - is there a different feature between LFC1 and LFC2 that should be highlighted?
  We prefer to keep these sentences separate as we are trying to contrast long-term and short-term variability with the SAM.

- L294 'We assume sea ice concentration anomalies are related to SST anomalies' - this could easily be checked.
  We have rephrased this section. Our initial motivation was to use an SST budget to highlight potential terms and mechanisms for SST variability and thus SIC variability. We did not aim to do a precise heat budget analysis. The new statements are in Lines 300-304.

- L361 'biases in Antarctic sea ice and global SST trends are likely related' - do the authors mean global or Southern Ocean? Unless I missed it, Wills et al. 2022 mostly focus on Southern Ocean SSTs.
  Wills et al., 2022 focuses on the large-scale SST pattern including the Southern Ocean and tropics. We have changed global to large-scale in the manuscript to not confuse the reader.

- L361 'It is still unclear whether global SST trends are the cause of or result of Southern Ocean trends' - I would change this to something a little more nuanced, as there is likely a two-way interaction.
  We discuss the two-way interaction right after this. We have refined the sentence to capture this (see Lines 370-380).

- L363 - This doesn't need to be implied - see e.g. Sadai et al. (2020, DOI: 10.1126/sci-adv.aaz1169) which shows that freshwater from the Antarctic ice sheet leads to Southern Ocean cooling and has a global impact.
  Thank you for this reference. We have added it and refined this sentence to acknowledge it does not need to be implied (374-376).

- L365 'Nonetheless' - I don't think this is the right conjunction
  We have changed this.

- Spatial plots - this is my personal preference, so the authors should feel free to disregard, but to me it is more intuitive to show sea ice gain as blue and loss as red
  We prefer to keep red as positive and blue as negative for all of the sea ice concentration anomaly figures, but we have changed Figure 1 to have the same colorbar as other sea ice concentration anomaly plots.